# Assessment of Colorectal Anastomosis with Intraoperative Colonoscopy: Its Role in Reducing Anastomotic Complications

**DOI:** 10.3390/biomedicines11041162

**Published:** 2023-04-12

**Authors:** Ri-Na Yoo, Ji-Yeon Mun, Hyeon-Min Cho, Bong-Hyeon Kye, Hyung-Jin Kim

**Affiliations:** 1Department of Surgery, St. Vincent’s Hospital, The Catholic University of Korea, Suwon 16247, Republic of Korea; ninayoo1111@gmail.com (R.-N.Y.); answl89@gmail.com (J.-Y.M.);; 2Department of Surgery, Eunpyeong St. Mary’s Hospital, The Catholic University of Korea, Seoul 03312, Republic of Korea

**Keywords:** colonoscopy, colorectal neoplasms, surgical anastomosis, anastomotic leak

## Abstract

The use of intraoperative colonoscopy (IOC) to evaluate the integrity of newly created anastomosis has been advocated by some surgeons. However, whether direct visualization of fresh anastomosis can help reduce anastomotic problems is still unclear. This study investigates the impact of immediate endoscopic assessment of colorectal anastomosis on anastomotic problems. This is a retrospective study conducted at a single center. Among six hundred forty-nine patients who underwent stapled anastomosis for left-sided colorectal cancer, the anastomotic complications were compared between patients who underwent IOC and those who did not. Additionally, patients with subsequent intervention after the IOC were compared to those without the intervention. Twenty-seven patients (5.0%) developed anastomotic leakage, and six (1.1%) experienced anastomotic bleeding postoperatively. Of the patients with IOC, 70 patients received reinforcement sutures to secure anastomotic stability. Of 70 patients, 39 patients showed abnormal findings in IOC. Thirty-seven patients (94.9%) who underwent reinforcement sutures did not develop postoperative anastomotic problems. This study demonstrates that IOC assessment with reinforcement sutures does not imminently reduce the rate of anastomotic complications. However, its use may play a role in detecting early technical failure and preventing postoperative anastomotic complications.

## 1. Introduction

Anastomotic complications in colorectal cancer surgery are the most feared and dreadful morbidity because of the disastrous impact on postoperative recovery and oncologic outcome. After left-sided colorectal cancer surgery, the leakage rate ranges from 2.6% to 12.3%, of which the overall anastomotic leakage rate is approximately 6% [1,2]. In addition, postoperative bleeding from stapled colorectal anastomosis has been reported to occur in up to 6.5% of patients [3]. Various risk factors in anastomotic problems, particularly leakage, include obesity, male sex, old age, advanced tumor stage, and preoperative chemoradiation, which are often difficult to modify [1]. Nevertheless, some factors are considered modifiable, particularly those associated with surgical procedures, such as excessive tension and poor perfusion on the anastomosis, bleeding in an anastomotic region, and the number of linear staples used [4].

To minimize anastomotic complications, surgeons have suggested evaluating the integrity of anastomosis intraoperatively [5,6]. Different intraoperative techniques are available to assess newly constructed anastomosis, such as the intraoperative air leakage test (IALT), intraoperative colonoscopy (IOC), and indocyanine green (ICG) test [7]. IALT is probably the handiest and most popular technique performed by surgeons, requiring little time and effort [8]. IOC allows the direct visualization of fresh anastomosis and immediate detection of disruption or bleeding at the anastomotic staple lines. It also directly insufflates the lumen with enough air pressure, allowing a simultaneous air leak test. In contrast, the ICG test evaluates the perfusion of the anastomosis by visualizing the fluorescence emitted by ICG under near-infrared light [9].

Each methodology’s effect on reducing anastomotic complications is inconclusive and still in the process of building evidence. Therefore, in this study, focusing on the role of intraoperative colonoscopy, we investigated its impact on lowering the anastomotic complications of leakage and bleeding.

## 2. Materials and Methods

Based on a prospectively collected database constructed from patient records, a retrospective review of patients who underwent operative management for colorectal cancer at our institution between January 2017 and December 2021 was conducted. The local ethics committee approved this study (IRB #VC22RISI0125). The requirement for informed consent was waived because of the retrospective design and because the analysis used anonymous clinical data and involved no additional procedure other than routine practices in a clinical setting, presenting no risk of harm to the patients.

The database consists of the following parameters: patient demographics, including age, sex, and body mass index (BMI), American Society of Anesthesiologists (ASA) class, preoperative condition of colonic obstruction or perforation, neoadjuvant therapy, type of surgery, operative approach, operation time, location of anastomosis (above or below the peritoneal reflection level), the height of anastomosis, type of anastomosis (stapled or hand-sewn), formation of diverting stoma, pathologic TNM stage, anastomotic leakage or bleeding, and diagnosis timing of anastomotic problem. Anastomotic leakage was defined and graded according to the definition of the International Study Group of Rectal Cancer [10].

### 2.1. Patients

Among 1210 consecutive patients, 1111 underwent primary anastomosis with or without diverting stoma (Figure 1). The exclusion criteria were patients who underwent non-sphincter saving surgery, local excision, bypass, or open and closed surgery. Of 1111 patients, 703 identified with left-sided colorectal cancer requiring anterior resection, low anterior resection, or intersphincteric resection were included in the study. The patients were categorized into two groups: stapled anastomosis and hand-sewn anastomosis. The patients with stapled anastomosis were further classified into two groups: those tested with IOC for the integrity of the anastomosis and those without the test. The primary outcome measure was the rate of anastomotic complications of bleeding and leakage. The anastomotic complication rates in the two subgroups were compared. The clinicopathologic and operative characteristics were evaluated and compared for the subgroups. Moreover, after IOC, the anastomotic complication rates were compared between patients who underwent an additional operative intervention, including transanal or transabdominal reinforcement sutures, and those who did not. After obtaining approval from the Institutional Review Board of St. Vincent Hospital at the Catholic University of Korea (VC22RISI0200), we retrospectively reviewed the patients’ data and clinical information.

### 2.2. Preoperative Condition and Management

Perioperative management for elective surgery was standardized as a clinical pathway system for all patients. Patients without signs and symptoms of colonic obstruction were given preoperative mechanical bowel preparation with 4 L of polyethylene glycol solution 24 h before the surgery. A single dose of second-generation cephalosporin was administered as a prophylactic antibiotic 30 min before the skin incision.

If colonic obstruction was present preoperatively, colonic decompression was carried out by colonoscopic insertion of a self-expandable metallic stent (SEMS) or the formation of a diverting stoma. When the obstruction was successfully relieved within 24 to 48 h, elective surgery was planned within 7 to 14 days. A diverting stoma was performed when the tumor was located within the pelvis and palpated on a digital rectal exam; SEMS was inserted for tumors located intraperitoneally. Emergency surgery was performed if a patient presented signs and symptoms of peritonitis and unstable hemodynamics with evidence of free air or impending perforation in a preoperative imaging study. During the operation, an individual surgeon decided whether to perform anastomosis with or without a diverting stoma.

Patients with cancer located within 15 cm from the anal verge, defined as rectal cancer, were subject to preoperative chemoradiotherapy (CRT) if the disease showed clinical stage (c) T3 or positive metastatic mesorectal node in imaging workups. Either short- or long-course CRT was given if indicated, depending on the patient’s preference and eligibility. The eligibility criteria include (1) histologically confirmed cancer; (2) distal tumor margin located within 8 cm from the anal verge; (3) cT3-4N0-2 classification determined by MRI and/or endorectal ultrasonography; (4) no evidence of distant metastasis; (5) Karnofsky performance score greater than 70; and (6) adequate bone marrow, liver, and renal functions (leukocyte count >4.0 × 10^9^/L, hemoglobin level > 10 g/dL, platelet count > 100 × 10^9^/L, serum bilirubin level <1.5 mg/dL, serum transaminase level < 2.5 times the normal upper limit, and serum creatinine level < 1.5 mg/dL) [11]. The long-course treatment consists of two cycles of intravenous 5-fluorouracil (5-FU) with a dose of 400 mg/m^2^ before radiotherapy and intravenous leucovorin with a dose of 20 mg/m^2^ before each dose of 5-FU on days 1–5 and 29–33 delivered concurrently with radiation of 45–50 Gy in 25–28 fractions to the pelvis. In the short-course treatment, capecitabine at a dose of 825 mg/m^2^ twice daily from days 1–12 was delivered concurrently with radiation of 33 Gy in 10 fractions for two weeks. Total mesorectal excision (TME) was performed 6 to 8 weeks after preoperative CRT.

### 2.3. Intraoperative Procedures

Three colorectal surgeons performed standardized surgical procedures for left-sided colorectal cancer, including high ligation of the inferior mesenteric artery (IMA) and splenic flexure mobilization. After resecting the specimens, intracorporeal end-to-end anastomosis was created by the double stapling method. If a tumor involved the distal rectum or anal canal, a transanal approach for intersphincteric resection was performed. An end-to-end anastomosis was created by the hand-sewn method. For the stapled anastomosis, the integrity of the anastomosis was double-checked—first, with an air leak test using a 50 cc enema syringe. Then, upon the availability of a colonoscopy, a flexible colonoscope was inserted through the anus and advanced to the stapler line. The anastomotic line and the proximal and distal limbs were evaluated for disruption, bleeding, and mucosal discoloration. During the air leak test and IOC assessment, the pelvis was filled with warm saline for the air bubbles from the anastomotic defect. The proximal limb above the anastomosis was clamped for air trapping in the colon. Loop ileostomy was constructed if (1) preoperative CRT was given, (2) complete total mesorectal excision was performed, or (3) the operator made a judgment call about the necessity of protecting the anastomosis.

### 2.4. Statistical Analysis

All statistical analyses were performed using IBM SPSS Statistics ver. 26.0 (IBM Corp., Armonk, NY, USA). Univariate analysis was performed using Fisher’s exact test or Pearson’s test for categorical variables and Student’s *t*-test for continuous variables. In addition, logistic regression analysis was carried out for multivariate analysis to evaluate the independent risk factors for anastomotic complications. *p* values less than 0.05 were considered statistically significant.

## 3. Results

Of 703 patients undergoing radical resection for left-sided colorectal cancer, 44 patients (6.3%) experienced anastomotic complications—38 (5.4%) with leakage and 6 (0.9%) with bleeding, as shown in Table 1. Anastomotic leakage mainly occurred in patients with low anterior resection (LAR) or intersphincteric resection (ISR). On the other hand, anastomotic bleeding occurred only in the patients undergoing anterior resection. An amount of 11.1% of patients with hand-sewn anastomosis experienced anastomotic leakage, while 5% of patients with stapled anastomosis experienced leakage. Anastomotic bleeding occurred at an overall rate of 0.9%, all in the stapled anastomosis. In those patients with stapled anastomosis, the bleeding occurred more frequently when the anastomosis was above 10 cm from the anal verge. On the contrary, the leakage was observed at the highest rate as the anastomosis was below 5 cm.

When the patients with anastomotic problems in stapled anastomosis were compared to those without these problems, a higher anastomotic complication rate was noticed in the male sex, in patients treated with neoadjuvant chemoradiotherapy and in patients with diverting stoma formation, as shown in Table 2. Additionally, lower anastomosis in the pelvis showed a higher rate of anastomotic complications. In the multivariate analysis, the male sex, neoadjuvant therapy, diverting stoma formation, and anastomosis located within 5 cm from the anal verge appeared to be significant risk factors for anastomotic complications.

In the patients with stapled anastomosis, the newly created anastomosis was assessed with IOC in 541 patients (83.4%). As shown in Table 3, the anastomotic leakage rates were similar in the patients with the IOC assessment compared to those without it. However, all anastomotic bleeding cases occurred in patients with IOC assessment. Furthermore, patients with IOC assessment experienced anastomotic complications not only within 30 days but also after 30 days. After the IOC assessment, seventy patients (12.9%) received the additional intraoperative intervention of reinforcement sutures due to anastomotic instability. As shown in Table 4, two patients with additional intraoperative intervention developed leakage within 30 days of the operation, but none had bleeding or leakage after 30 days. On the other hand, of the patients without additional intervention, 31 patients experienced anastomotic complications: 5.3% experienced leakage, and 1.3% experienced bleeding. The anastomotic complications occurred mainly within postoperative day 30; however, four patients developed leakage after postoperative day 30.

Table 5 details the abnormalities observed during the IOC assessment and the underlying reasons for additional intraoperative intervention. The most common cause was bleeding or hematoma formation in the stapler line (53.9%). The others include mucosal edema of the proximal limb and stapler line disruption with or without air leakage. The two patients who developed leakage after the reinforcement sutures showed abnormal findings in the IOC assessment—one with proximal bowel limb edema and bleeding and the other with air leakage. Both patients required reoperation within 14 days of the index surgery—one underwent Hartmann’s procedure due to the proximal limb ischemia, and the other underwent resection of the proximal limb and recreation of the anastomosis due to anastomotic failure.

Table 6 shows the clinical, surgical, and pathological characteristics of patients who were more likely to have undergone additional intraoperative intervention. Patients with colonic obstruction were more likely to undergo additional intervention. Patients who underwent emergency, open surgery, or surgery requiring a long operation time underwent additional intervention. Additionally, a higher proportion of the patients who underwent anterior resection received additional intervention than those who underwent low anterior resection. Patients with an advanced tumor stage were more likely to receive the additional reinforcement suture in the anastomosis.

## 4. Discussion

For the early diagnosis and intervention of anastomotic complications, the use of IOC to assess the integrity of anastomosis has been employed by surgeons worldwide. However, the effectiveness of IOC in reducing anastomotic complications has been controversial and still lacks sufficient evidence. In a systematic review of different intraoperative assessment techniques, Hirst et al. reported that the IOC assessment might provide potential advantages in detecting anatomical abnormalities; however, the impact on reducing anastomotic leakage failed to reach clinical significance [12]. The more recent meta-analysis by Kryzauskas et al., including 12 primary studies, suggested that IOC could reduce the anastomotic leakage rate following lower gastrointestinal tract resection [6]. However, the authors doubted the efficacy of IOC based on the controversial results of the two studies. Because those two studies included a small number of patients, the impact of the pooled analysis on the study result was only minor. In addition, only two randomized controlled trials (RCTs) are available in the literature. Beard et al. and Ivanov et al. demonstrated that the IOC could effectively lower the anastomotic leakage rate [13,14]. However, concerns about the small sample size and the outdated data on surgical technology and techniques hinder a firm conclusion.

Nevertheless, in this study, including a relatively homogenous cohort with a larger sample size than in other studies, the anastomotic leakage rate was similar in either group of patients with or without IOC—5% vs. 4.7%. Postoperative anastomotic bleeding is a noticeable complication that occurred after the IOC assessment. Six patients who showed no abnormalities during the IOC assessment developed anastomotic bleeding within postoperative Day 2, and they were all managed by endoscopic clipping successfully without developing further complications. Consistent with these results, Shibuya et al. and Shamiyeh et al. reported that postoperative anastomotic bleeding and leakage occurred in patients who did not show any abnormal findings during IOC assessment [15,16]. Caused by submucosal edema and ischemia during the first 24 to 48 h, local inflammation and tissue necrosis of anastomosis can induce postoperative anastomotic bleeding in the process of wound healing [17]. Such evidence suggests that the IOC assessment does not readily predict anastomotic complications.

However, most patients who showed abnormalities in the IOC assessment could avoid anastomotic complications after the additional intervention of reinforcement sutures. In this study, 94.9% of patients with unstable anastomosis detected intraoperatively could avoid postoperative anastomotic complications. Air leakage detected intraoperatively in four out of five patients could be successfully managed by immediate intervention. Other anastomotic problems, particularly bleeding, could be treated, resulting in no further postoperative bleeding. Several other studies also demonstrated that technical anastomosis failure was readily detected in IOC and managed [13,14,15]. These results suggest that the routine use of IOC can help reduce anastomotic complications caused by possible technical errors, such as mucosal edema, bleeding, staple line disruption, and air leakage, in the IOC assessment indicating prompt intervention and management.

On the other hand, even if the additional intervention of reinforcement suture was performed due to abnormal findings detected in IOC, anastomotic leakage in some patients was not controlled. Li et al. reported two patients with postoperative leakage after the IOC testing showing no defect and a negative air leakage test [18]. Additionally, Lanthaler et al. reported a similar observation of patients developing postoperative leakage that was not detected in intraoperative testing [19]. The authors suggested that the causes of postoperative anastomotic insufficiency could be inadequate anastomosis blood supply or too much tension. Consistent with the previous reports, two patients in this study required reoperation for anastomotic leakage due to proximal limb ischemia. Both patients eventually required reoperation—one with resection of the proximal limb and end ileostomy and the other with the re-creation of the anastomosis with a temporary stoma. Since high ligation of the IMA was performed in these patients, the insufficient blood supply to the proximal colonic limb might have progressed and aggravated ischemia. In addition, a retrospective review of the patient who underwent resection of the proximal limb and end ileostomy revealed that the patient had suffered from a cerebrovascular accident. Thus, pre-existing vascular insufficiency probably affected mesenteric artery ischemia; consequently, the patient ended up with resection of the remnant colon and end ileostomy. Inadvertent compromise of vascular arcades might have been caused by ligation of the left colic artery, resulting in nonocclusive ischemia [20,21,22]. Identifying the mucosal discoloration and the proximal limb ischemia might be challenging on IOC testing during the operation. Reinforcement suture intervention is undoubtedly inadequate to secure anastomosis and correct the insufficient blood flow. Nonetheless, ICG testing may be an excellent additional method to assess the actual perfusion of the new anastomosis and assess vascular compromise [7].

The etiology of anastomotic complications, particularly leakage, is multifactorial, in which the risk factors are mainly categorized into three different aspects—patient-, tumor-, and surgical procedure-related. Patient- and tumor-related factors are often difficult to change or non-modifiable. However, preventive measures, such as splenic flexure mobilization for tension-free anastomosis or a temporary stoma formation, can be adopted in surgical procedures to decrease the rate and the associated morbidity of anastomotic complications [2,23]. In this study, patients with LAR were indicated for temporary stoma if neoadjuvant chemoradiotherapy was given, the anastomosis level within 8 cm from the anal verge, or unstable anastomosis with abnormality found in the IOC assessment. In addition to the splenic flexure mobilization and diverting stoma formation, the IOC assessment was performed routinely to reduce the anastomotic complication rates in our institute. As a result, postoperative anastomotic complications could be prevented with prompt treatment in 94.9% of patients with technical errors detected in IOC. Although not all complications are detected and predictable, the routine use of IOC may be valuable for evaluating technical failures that are immediately treatable. Reflecting that the anastomosis located below 5 cm from the anal verge poses a significant risk of leakage, patients with low anastomosis would require IOC testing and the subsequent intervention of reinforcement suture as a preventive measure. Further evaluation of possible technical errors not identified in the present study may help expand the indications of intraoperative intervention, particularly for patients with multiple risk factors.

Despite full splenic flexure mobilization and diverting stoma formation, anastomotic leakage occurred after postoperative Day 30 in four patients who did not show abnormalities in the IOC assessment. Three patients developed minor leakage symptoms on the postoperative Days 5 to 7 and were managed conservatively. One patient did not show any signs or symptoms of pelvic sepsis. However, they developed major signs and symptoms of anastomotic leakage at least 8 to 12 weeks after the closure of diverting stoma. All were male patients who underwent neoadjuvant chemoradiotherapy for locally advanced rectal cancer. Consistent with our multivariate analysis of risk factors for leakage in the stapled anastomosis, all three components are considered significant risk factors for anastomotic leakage [2,24,25]. Regardless of various preventive measures, including preoperative mechanical bowel preparation, splenic flexure mobilization, formation of diverting stoma, and IOC assessment, anastomotic leakage was unavoidable in those patients. This finding suggests that along with ischemia, other factors, such as gut microbiota or remnant tumor cells, may influence the intestinal anastomosis healing process [26,27]. Further investigation on the role of gut microbiota in intestinal wound healing is necessary, which may help identify other modifiable factors related to anastomotic leakage.

This study has some limitations. First, it was a retrospective study conducted at a single institution. Nevertheless, this study included a relatively homogenous, large cohort of colorectal cancer patients evaluated with IOC for newly created anastomosis. In addition, the database was prospectively collected without many missing data. Second, there could be inter-surgeon variability in the interpretation of endoscopic findings, which might influence the study outcome. However, the perioperative management and surgical procedures were all standardized, and all three colorectal surgeons strictly adhered to the standard protocol. In addition, they are all board-certified in colorectal surgery and colonoscopy; thus, endoscopic findings and descriptions may not differ much among them.

## 5. Conclusions

The IOC assessment has the core value of confirming anastomosis integrity by directly visualizing newly created anastomosis. Indeed, the routine use of IOC may benefit patients with unstable anastomosis detected in IOC, allowing immediate intervention. Quick assessment with IOC may reduce technical error, aiding in securing anastomosis of patients with multiple risk factors. However, IOC testing is limited in predicting anastomotic insufficiency caused by progressive ischemia or other causes interfering with wound healing. Further investigation on combining other testing methods evaluating vascularity and perfusion, such as ICG, with IOC testing is necessary. Perhaps applying both tests may help further reduce the anastomotic complication to achieve optimal and secure anastomosis.

## Figures and Tables

**Figure 1 biomedicines-11-01162-f001:**
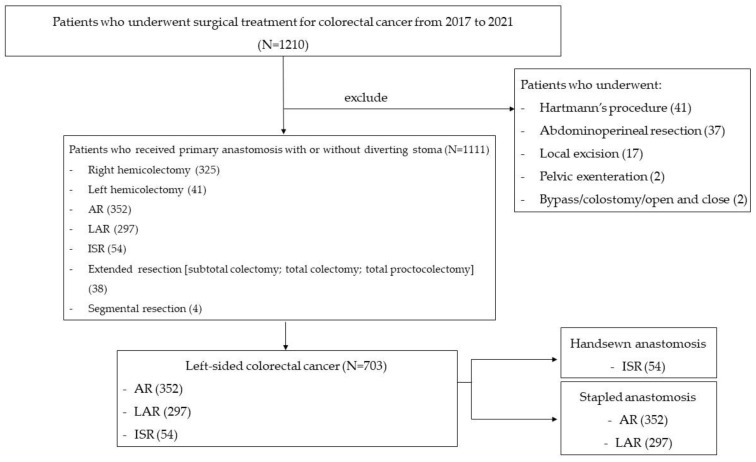
Patients flow chart. Abbreviations: AR, anterior resection; LAR, low anterior resection; ISR, intersphincteric resection.

**Table 1 biomedicines-11-01162-t001:** Rate of anastomotic complications for left-sided colorectal cancer.

Total *n* = 703		No Complication (*n* = 659, 93.7%)	Leakage (*n* = 38, 5.4%)	Bleeding (*n* = 6, 0.9%)	*p*-Value
AR (*n* = 351)		342 (97.4%)	3 (0.9%)	6 (1.7%)	<0.001
LAR (*n* = 298)		269 (90.3%)	29 (9.7%)	0 (0%)	
ISR (*n* = 54)		48 (88.9%)	6 (11.1%)	0 (0%)	
Anastomosis type	Handsewn (*n* = 54)	48 (88.9%)	6 (11.1%)	0 (0%)	0.160
	Stapled (*n* = 649)	611 (94.1%)	32 (5.0%)	6 (0.9%)	
Stapled anastomosis (*n* = 649)		No problem (*n* = 611)	Leakage (*n* = 32)	Bleeding (*n* = 6)	*p*-value
Height of the anastomosis from the anal verge	>10 cm	239 (97.2%)	2 (0.8%)	5 (2.0%)	<0.001
	>5~10 cm	116 (93.5%)	7 (5.6%)	1 (0.8%)	
	≤5 cm	153 (89.5%)	18 (10.5%)	0 (0%)	

AR, anterior resection; LAR, low anterior resection; ISR, intersphincteric resection.

**Table 2 biomedicines-11-01162-t002:** Univariate and multivariate analysis comparing patients with and without anastomotic complications in stapled anastomosis.

Univariate Analysis					Multivariate Analysis		
Stapled Anastomosis (*n* = 649)		No Complications (*n* = 611)	Leakage or Bleeding (*n* = 38)	*p*-Value	Odds Ratio	95% Confidence Interval	*p*-Value
Patient characteristics							
Age (Mean ± SD)		64.75 ± 11.52	63.05 ± 9.86	0.129	0.98	0.944–1.018	0.308
	<65	316 (51.7%)	21 (55.3%)	0.739			
	≥65	295 (48.3%)	17 (44.7%)				
Sex	Male	377 (61.7%)	33 (86.8%)	0.002	10.02	2.29–43.96	0.002
	Female	234 (38.3%)	5 (13.2%)		1		
BMI (Mean ± SD, kg/m^2^)		24.47 ± 3.44	24.30 ± 3.29	0.742			
ASA	I/II	557 (91.2%)	36 (94.7%)	0.763			
	III/IV	54 (8.8%)	2 (5.3%)				
Preoperative disease status							
Colonic perforation before surgery	No	603 (94.1%)	38 (5.9%)	1.0			
	Yes	8 (100%)	0 (0.0%)				
Colonic obstruction before surgery	No	524 (94.2%)	32 (5.8%)	0.811			
	Yes	87 (93.5%)	6 (6.5%)				
Colonic obstruction requiring preoperative decompression	No	571 (93.5%)	35 (92.1%)	0.723			
	Diversion	7 (1.1%)	1 (2.6%)				
	Metallic Stent	33 (5.4%)	2 (5.3%)				
Neoadjuvant therapy	Yes	123 (20.1%)	19 (13.4%)	<0.001	4.26	1.08–19.88	0.039
	No	488 (79.9%)	19 (3.7%)		1		
Neoadjuvant therapy-yes	None	488 (79.9%)	19 (50.0%)	<0.001			
	CTx + RTx	107 (17.5%)	17 (44.7%)				
	Cytotoxic CTx only	2 (0.3%)	0 (0%)				
	Cytotoxic CTx + target agent	12 (2%)	2 (5.3%)				
	Cytotoxic CTx + target agent + RTx	2 (0.3%)	0 (0%)				
Operation characteristics							
Emergency surgery	No	606 (94.2%)	37 (5.8%)	0.305			
	Yes	5 (83.3%)	1 (16.7%)				
Surgical approach	Open	64 (97.0%)	2 (5.3%)	0.413			
	Minimally invasive	547 (89.5%)	36 (6.2%)				
Operation time (Mean ± SD, min)		217.26 ± 75.12	252.08 ± 81.72	0.192	1.00	0.995–1.007	0.672
IOC	No	103 (16.9%)	5 (13.2%)	0.659			
	Yes	508 (83.1%)	33 (86.8%)				
Additional intraoperative intervention	No	532 (87.1%)	35 (92.1%)	0.459			
	Yes	79 (12.9%)	3 (7.9%)				
Formation of diverting stoma	No	406 (66.4%)	15 (39.5%)	0.001	1		
	Yes	205 (33.6%)	23 (60.5%)		7.88	1.306–47.57	0.024
Pathologic characteristics							
Pathologic T stage	p or yp CR or T1	135 (22.1%)	5 (13.2%)	0.101			
	p or yp T2	93 (15.2%)	11 (28.9%)				
	p or yp T3	285 (46.6%)	18 (47.4%)				
	p or yp T4	98 (16.0%)	4 (10.5%)				
Circumferential margin status	Negative	594 (97.2%)	37 (97.4%)	1.0			
	Positive	17 (2.8%)	1 (2.6%)				
Pathologic TNM stage	Stage 0 or I	195 (31.9%)	15 (39.5%)	0.239			
	Stage II	162 (26.5%)	11 (28.9%)				
	Stage III	204 (33.4%)	7 (18.4%)				
	Stage IV	50 (8.2%)	5 (13.2%)				
Height of the anastomosis (Mean ± SD, cm)		9.46 ± 5.48	6.21 ± 4.0	0.005	0.88	0.72–1.07	0.165
Height of the anastomosis from the anal verge	>10 cm	297 (48.6%)	7 (18.4%)	<0.001	1		
	>5~10 cm	135 (22.1%)	8 (21.1%)		2.92	0.968–8.792	0.057
	≤5 cm	179 (29.3%)	23 (60.5%)		6.77	1.56–29.31	0.011

SD, standard deviation; BMI, body mass index; ASA, American Society of Anesthesiologists; CTx, Chemotherapy; RTx, Radiotherapy; IOC, Intraoperative colonoscopy; CR, Complete response.

**Table 3 biomedicines-11-01162-t003:** Comparison of anastomotic complication rates based on the intraoperative colonoscopic evaluation.

Stapled Anastomosis (*n* = 649)		IOC (−) (*n* = 108, 16.6%)	IOC (+) (*n* = 541, 83.4%)	*p*-Value
Anastomotic complications	None	103 (95.4%)	508 (93.9%)	0.537
	Leakage	5 (4.6%)	27 (5.0%)	
	Bleeding	0 (0%)	6 (1.1%)	
Anastomotic complications	None	103 (95.4%)	508 (93.9%)	0.634
	Within 30 days	5 (4.6%)	29 (5.4%)	
	After 30 days	0 (0%)	4 (0.7%)	

IOC, intraoperative colonoscopy.

**Table 4 biomedicines-11-01162-t004:** Comparison of anastomotic complication rates based on the additional intraoperative intervention.

IOC (+) (*n*= 541)		Additional Intraoperative Intervention (−) (*n* = 471, 87.1%)	Additional Intraoperative Intervention (+) (*n* = 70, 12.9%)	*p* Value
Anastomotic complications	None	440 (93.4%)	68 (97.1%)	0.437
	Leakage	25 (5.3%)	2 (2.9%)	
	Bleeding	6 (1.3%)	0 (0%)	
Anastomotic complications	None	440 (93.4%)	68 (97.1%)	0.444
	Within 30 days	27 (5.7%)	2 (2.9%)	
	After 30 days	4 (0.8%)	0 (0%)	

IOC, intraoperative colonoscopy.

**Table 5 biomedicines-11-01162-t005:** IOC findings of intraoperative anastomotic complications and postoperative anastomotic complications.

Additional Intraoperative Intervention (+) (*n* = 70)		Abnormal Finding (−) in IOC (*n* = 31, 44.3%)	Abnormal Finding (+) in IOC (*n* = 39, 55.7%)
Intraoperative anastomotic complications	Mucosal edema		5 (12.8%)
	Stapler disruption		8 (20.5%)
	Bleeding or hematoma		21 (53.9%)
	Air leakage		5 (12.8%)
Postoperative anastomotic complications	None	31 (100%)	37 (94.9%)
	Leakage	0 (0%)	2 (5.1%)
	Bleeding	0 (0%)	0 (0%)
Postoperative anastomotic complications	None	31 (100%)	37 (94.9%)
	Within 30 days	0 (0%)	2 (5.1%)
	After 30 days	0 (0%)	0 (0%)

IOC, intraoperative colonoscopy.

**Table 6 biomedicines-11-01162-t006:** Univariate analysis comparing patients with and without additional intraoperative intervention after IOC evaluation.

Patients with IOC (*n* = 541)		Additional Intraoperative Intervention (−) (*n* = 471)	Additional Intraoperative Intervention (+) (*n*= 70)	*p* Value
Patient characteristics				
Age (Mean ± SD)		64.75 ± 10.89	64.32 ± 13.54	0.765
	<65	247 (87.0%)	37 (13%)	1.0
	≥65	224 (87.4%)	33 (12.8%)	
Sex	Male	292 (86.1%)	47 (13.9%)	0.430
	Female	179 (88.6%)	23 (11.4%)	
BMI (Mean ± SD, kg/m^2^)		24.54 ± 3.30	23.92 ± 3.61	0.150
ASA	I/II	434 (87.1%)	64 (12.9%)	0.813
	III/IV	37 (86.0%)	6 (14.0%)	
Preoperative disease status				
Colonic perforation before surgery	No	467 (87.1%)	69 (12.9%)	0.501
	Yes	4 (80.0%)	1 (20.0%)	
Colonic obstruction before surgery	No	421 (89.6%)	49 (10.4%)	<0.001
	Yes	50 (70.4%)	21 (29.6%)	
Colonic obstruction requiring preoperative decompression	No	452 (89.2%)	56 (11.0%)	<0.001
	Diversion	5 (71.4%)	2 (28.6%)	
	Metallic Stent	14 (53.8%)	12 (46.2%)	
Neoadjuvant therapy	No	110 (92.4%)	9 (7.6%)	0.062
	Yes	361 (85.5%)	61 (14.5%)	
Operation characteristics				
Emergency surgery	No	470 (87.4%)	68 (12.6%)	0.045
	Yes	1 (33.3%)	2 (66.7%)	
Surgical approach	Open	22 (47.8%)	24 (52.2%)	<0.001
	Minimally invasive	449 (90.7%)	46 (9.3%)	
Operation time (Mean ± SD, min)		212.29 ± 68.45	235.52 ± 100.26	0.014
Surgery type	AR	240 (81.9%)	53 (18.1%)	<0.001
	LAR	232 (93.5%)	16 (6.5%)	
Formation of diverting stoma	No	301 (85.0%)	53 (15.0%)	0.059
	Yes	170 (91.4%)	17 (9.1%)	
Anastomotic complications (bleeding and leakage)	No	440 (86.6%)	68 (13.4%)	0.292
	Yes	31 (93.9%)	2 (6.1%)	
Pathologic characteristics				
Pathologic T stage	p or yp CR or T1	111 (94.1%)	7 (5.9%)	<0.001
	p or yp T2	78 (90.7%)	8 (9.3%)	
	p or yp T3	225 (87.9%)	31 (12.1%)	
	p or yp T4	57 (70.4%)	24 (29.6%)	
Circumferential margin status	Negative	461 (87.1%)	68 (12.9%)	0.660
	Positive	10 (2.1%)	2 (16.7%)	
Pathologic TNM stage	Stage 0 or I	167 (93.3%)	12 (6.7%)	0.009
	Stage II	124 (82.7%)	26 (17.3%)	
	Stage III	147 (86.5%)	23 (13.5%)	
	Stage IV	33 (78.6%)	9 (21.4%)	
Anastomosis Level				
Height of the anastomosis (Mean ± SD, cm)		9.05 ± 5.47	11.12 ± 4.91	0.012
Height of the anastomosis from the anal verge	>10 cm	206 (43.7%)	40 (57.1%)	0.033
	>5~10 cm	107 (22.7%)	17 (24.3%)	
	≤5 cm	158 (33.5%)	13 (18.6%)	

SD, Standard deviation; BMI, body mass index; ASA, American Society of Anesthesiologists; IOC, intraoperative colonoscopy.

## Data Availability

MDPI Research Data Policies at https://www.mdpi.com/ethics (accesed on 19 March 2023).

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
