# Peer review of "Assessment of Colorectal Anastomosis with Intraoperative Colonoscopy: Its Role in Reducing Anastomotic Complications"

_biomedicines, 2023, doi:10.3390/biomedicines11041162_

Round 1

Reviewer 1 Report

In this paper the Authors presented the evaluation of colorectal anastomosis with intraoperative colonoscopy and its role anastomotic complications. It is a not so common event. A comprehensive and extensive literature review of the NCBI database PubMed was also carried out. The article was well conducted and it is interesting in its fields. It is a well-structured paper, written in good English and the References are up dated. 

Minor issues:

The anastomotic complications are among the most frequent sequelae after colon cancer surgery. Even at long time an important evaluation is the possibility to have an anastomotic recurrence.

Therefore, please consider:

 “Isolated repeated anastomotic recurrence after sigmoidectomy. Conzo G, Mauriello C, Gambardella C, Cavallo F, Tartaglia E, Napolitano S, Santini L. World J Gastroenterol. 2014 Nov 21;20(43):16343-8..”

Author Response

Thank you very much for your valuable comment. 

Reading through the reference you mentioned, I noticed that intraoperative colonoscopy might help confirm adequate resection margin. I added the reference along with the other factors that interfere with the anastomosis healing process.  

Reviewer 2 Report

The results cannot be analysed because the patients are not comparable.I suggest that the authors rework the methodology by carrying out either a case control study or at best a study using a propensity score 

Author Response

Thank you for your valuable comment.

As you recommended, we ran 1:1 propensity score matching (PSM) based on age, BMI, and ASA class. 

Using the PSM cohort, univariate analysis was performed.

Univariate analysis showed that anastomotic complication was not different when patients with the intraoperative colonoscopic test to those without.  

Also, the intraoperative additional intervention did not make any difference in complication rates.

The authors think that because the rate of anastomotic complication is less than 10%, it would be important to include all the anastomotic complication cases instead of selecting patients based on matching.  

We added the additional statistical result in the attached file. 

Round 2

Reviewer 2 Report

The authors responded point by point to questions and comments